# Parents, Teachers, and Community: A Team Approach to Developing Physical Competence in Children

**DOI:** 10.3390/children10081364

**Published:** 2023-08-09

**Authors:** Matthew S. Chapelski, Marta C. Erlandson, Alexandra L. Stoddart, Amanda Froehlich Chow, Adam D. G. Baxter-Jones, M. Louise Humbert

**Affiliations:** 1College of Kinesiology, University of Saskatchewan, Saskatoon, SK S7N 5B2, Canada; m.chapelski@usask.ca (M.S.C.); marta.erlandson@usask.ca (M.C.E.); baxter.jones@usask.ca (A.D.G.B.-J.); 2Health, Outdoor, and Physical Education, Faculty of Education, University of Regina, Regina, SK S4S 0A2, Canada; alexandra.stoddart@uregina.ca; 3School of Public Health, University of Saskatchewan, Saskatoon, SK S7N 5E5, Canada; a.fc@usask.ca

**Keywords:** physical competence, young children, physical literacy intervention, physical activity

## Abstract

Increasing children’s physical activity engagement has short- and long-term health benefits. Developing physical competence is a key component of children’s engagement in physical activity. The purpose of our study was to assess if a 12-week home, school, and community-based physical literacy intervention improved the physical competence of children in kindergarten and grade one. Four schools were either assigned to receive the intervention (n = 2 schools) or continue with their usual practice (control sites) (n = 2 schools). Physical competence was evaluated pre- and post-intervention in 103 intervention (41 female) and 83 usual practice (36 female) children using PLAY Fun. PLAY Parent and PLAY Coach tools measured parent and teacher perspectives of children’s physical competence, respectively. The intervention effect was assessed with repeated measures MANOVA to evaluate change in physical competence, with alpha set at *p* < 0.05. Children in both groups improved their locomotor, object control, and overall physical competence (*p* < 0.05) over the 12-week intervention. There was a significant intervention effect for locomotor and overall physical competence (*p* < 0.05). Interestingly, parents did not perceive these changes in physical competence (*p* > 0.05). However, teachers perceived improved physical competence for children in the intervention. Our physical literacy intervention improved the physical competence of children in kindergarten and grade one.

## 1. Introduction

Physical activity is beneficial to the health and wellness of children [1,2,3], and these benefits track into adulthood [2]. Despite the known benefits, only 17.5% of Canadian children 5 to 11 years of age were found to be reaching the Canadian physical activity guideline of 60 min of moderate-to-vigorous physical activity per day in 2020 [4]. In addition, the COVID-19 pandemic has been suggested to have further negatively impacted physical activity levels in children and youth [5,6,7]. Therefore, finding ways to improve physical activity engagement is imperative to children’s health throughout their lifetime.

One way to improve physical activity engagement is through the development of physical literacy. Physical literacy is defined as “the motivation, confidence, physical competence, knowledge, and understanding to value and take responsibility for engagement in physical activities for life” [8]. Components of physical literacy, such as confidence [9,10] and motivation [11], have been found to impact physical activity engagement. However, the positive relationship between physical competence and physical activity participation is the most well-established [11,12,13,14]. Moreover, De Meester and colleagues [10] identified a potential proficiency barrier between physical competence and physical activity engagement, where children 9 years of age with high physical competence were approximately three times more likely to reach the physical activity guidelines. While the development of physical literacy is essential for the long-term physical activity engagement of everyone, focusing on improving physical competence in the early years of life might have the greatest impact on children’s physical activity engagement [11,15,16].

Children 7 years of age with low physical competence continue to have low physical competence 5 years later compared to their peers [17]. These differences in physical competence impact a child’s daily physical activity. De Meester et al. found that 10% of children with low physical competence met the physical activity guidelines, compared to 25% of children with average physical competence, and 41% of children with high physical competence [10]. Thus, the development of physical competence is a crucial first step to improving physical activity engagement in children.

While our intuition may lead us to believe sport participation or time spent in sport are the most important, Engström [18] found that neither participation in nor time spent in sport was a significant predictor of long-term physical activity participation; however, they did find that sports breadth was a significant predictor. In other words, children who can perform a wide array of motor skills are more likely to be active as they age [18]. As a result, children who engage in a wide variety of motor skills opportunities in different settings (e.g., home, school, and community) and environments (e.g., ice, snow, and water) [18,19,20,21,22,23] may have greater improvements in physical competence. This is highlighted in a study that found Serbian children participating in a 1-year multisport program two times a week had greater physical competence than children only involved in soccer [22].

Despite the importance of sport, 26% of Canadian children 5 to 17 years of age do not participate in organized sport [4], so unstructured physical activity or free play is another way to develop physical competence. Free play can happen in any environment or setting, with or without peers, uses various types of equipment, and requires no planning, which allows children to have autonomy over the development of their physical competence. Free play has a positive influence on the development of locomotor physical competence and engagement in physical activity [23]. However, its’ impact on object control and physical competence is limited by access to equipment [23]. Additionally, a review found that free play has the lowest impact on developing physical competence compared to structured opportunities like physical education classes and motor development interventions [24].

For children in kindergarten and grade one, two of the most important settings that provide opportunities for the development of physical competence are the home and school. These settings are crucial since the family has the greatest and longest impact on a child’s behavior towards physical activity [25]. Furthermore, the quality of parental engagement is more impactful than the quantity of their involvement [25]. A survey found 88% of parents felt they had the greatest responsibility for the development of their child’s physical literacy [26], which included physical competence. The school is also crucial, as children spend the majority of their time in this setting, and many school divisions have physical education mandates [27]. However, while physical competence is included in many physical education curricula [27,28], a study in Saskatchewan found that only 60% of teachers reported that they felt the outcomes in the provincial curriculum were beneficial in addressing physical literacy development [29]. Some reasons the curriculum was not beneficial included vagueness about the concept of physical literacy, only the basics for physical literacy development being provided, and beliefs that the curriculum was outdated [29]. Also, the curriculum is unable to address the lack of proper equipment or time needed to develop physical literacy and competence [29]. Therefore, while the school and home may be favorable settings to develop physical literacy and, in particular, physical competence, resources may be lacking to support parents and teachers in this endeavor.

Interventions in schools focused on improving physical competence have yielded improvements after the intervention [30,31,32,33] as well as when compared to usual practice sites [31,32,33]. Interventions in the community setting have also found improvements in physical competence compared to baseline [34,35,36,37] and greater physical competence than children in the control group [35,37]. However, to our knowledge, no studies have examined the potential of a multi-setting physical literacy intervention to improve physical competence. Therefore, the purpose of this study was to assess the effect of a 12-week home, school, and community-based physical literacy intervention on the physical competence of children in kindergarten and grade one. We hypothesized that children who participated in the intervention would have higher levels of physical competence than those exposed to usual practice. Although this study implemented an intervention with a physical literacy-enriched focus, due to the age level of the participants and constraints of the available measurement tools at the time the study was conducted (2016), only physical competence results will be discussed.

## 2. Materials and Methods

### 2.1. Research Design and Participants

This study assessed the effect of a 12-week physical literacy intervention on the physical competence of children in kindergarten and grade one using a repeated measures cluster randomized control trial. Two hundred and seventeen (n = 118 intervention and n = 99 usual practice) kindergarten and grade one students were recruited from two cities. In each city, the local Catholic School Divisions selected one school to act as an intervention site and another school to continue their usual practice (control site). In total four schools, matched for neighborhood demographics (such as socioeconomic status, presence of physical education specialists, and size), participated in the project (n = 2 intervention schools and n = 2 usual practice schools). The study received ethical approval from the University of Saskatchewan’s Behavioural Ethics Review board (BEH#16-66) and permission from the participating Catholic School Divisions. In one city, the participating school division requested that the research team use passive consent. Thus, children were automatically involved in the study unless their parents chose to opt out. In the other city, parents provided written informed consent; all children at both sites provided verbal assent prior to data collection.

### 2.2. Physical Competence Assessment

Physical competence was assessed using the Physical Literacy Assessment for Youth (PLAY) Tools: specifically, PLAY Fun, PLAY Parent, and PLAY Coach. PLAY Fun assessed running, locomotor, object control, and balance skills [38]. PLAY Parent evaluated the parent’s perceptions of their child’s locomotor and object control physical competence [39], and PLAY Coach was used to appraise the teacher’s perceptions of their students’ balance, object control, locomotor, and overall physical competence [40]. PLAY Fun [41,42,43,44] and PLAY Parent [43] are valid measures of physical competence and have a moderate relationship [43]. The validity and reliability of PLAY Coach have not yet been studied. PLAY Fun evaluated the intervention’s impact on children’s physical competence, while PLAY Parent and PLAY Coach assessed if parents and teachers perceived any change in the children’s physical competence after the intervention.

Since PLAY Fun was designed for children aged 7 and up [38], modifications occurred to make the tool age-appropriate. When this study took place (2016), there were limited physical literacy assessment tools, so we decided to alter PLAY Fun. We removed the skills run jump then land on two feet, crossovers, strike with stick, hand dribble, foot dribble, drop to the ground, and lift and lower since they were not age-appropriate and did not align with the Saskatchewan physical education curriculum for kindergarten and grade one students [27]. Additionally, one-handed catch was altered to two-handed catch, and balance walk was adapted to stationary balance. This adaptation allowed the research team to ensure that the tasks were developmentally appropriate. The tool was not validated after the alterations. However, only the criteria for balance was altered from the original version created by Canadian Sport for Life [38] to account for the stationary balance test. The stationary balance test utilized the criteria developed for the physical literacy passport for life assessment tool [45].

In order to facilitate the PLAY Fun assessment, a team of assessors (n = 5) was formed consisting of researchers and graduate students experienced in movement analysis with children to properly apply the PLAY Fun criterion to motor skills. Assessors were trained by a leader in physical literacy assessment (ALS) and practiced by watching a film of PLAY Fun skills and assessing them based on the criteria with feedback from the facilitator. The study’s data collection followed the recommended PLAY Fun procedures [38]. For example, one assessor asked a participant to perform a skill from a script, such as, “I want you to run a square around the pylons. I want you to run a square as best you can. Ready? Run now” for the run a square skill [38]. The child then performed the skill and was given a score based on the grading criteria developed by Canadian Sport for Life (scores range from 0 to 100). This was repeated for all 10 skills. An overall physical competence score was then calculated by dividing the sum of the 10 skills by 10. Scores were divided into physical competence stages of initial (score 0–25), emerging (26–50), competent (51–75), and proficient (76–100) [39]. Furthermore, skills were divided into running (run a square and run there and back), locomotor (skip, gallop, hop, and jump), object control (throw, catch, and kick), and balance domains. In total, there were five assessors who tested the same two skills pre- and post-intervention.

### 2.3. Intervention

The 12-week intervention targeted home, school, and community settings. At the home level, children were provided with a backpack that contained equipment and instructional cards or physical literacy practice at home (Appendix A). The activities and accompanying equipment in these backpacks were changed weekly. Instructional cards were designed to focus on developing confidence, competence, and motivation for a broad range of skills in both indoor and outdoor spaces in both winter and summer environments. The instructional cards included information on how to perform the activity, cues to provide the children, and the skills that were being developed, as well as questions for parents to ask their children to help build their confidence and competence with the movement skills. Children kept all equipment and brought their backpack back to school each week to have new activities and equipment inserted into their backpack. Each week, the classroom teacher encouraged the children to use the items with their families. All children in the participating classroom received the backpacks; however, only those students with parental consent were assessed.

Twenty grade-specific physical literacy-focused lesson plans were developed by a physical education specialist to be taught during physical education classes (Appendix A). The lesson plans displayed the curricular outcome that would be achieved with the session as well as a detailed explanation of 3–5 activities that would develop the motor skills being addressed, along with the equipment needed, and cues for teachers. Two experienced facilitators taught two curriculum-based physical literacy lesson plans during physical education classes each week for the 12-week period. The classroom teachers, who were generalists, observed the instruction and often joined in the activities with their students. The classroom teacher then was asked to reteach one of the lessons in a third physical education class that week. Each facilitator worked alongside the classroom teacher, providing embedded professional development by teaching each lesson with the classroom teacher assisting. Similar to the home-based portion, every child in each of the classrooms participated in the physical literacy-focused lessons during physical education classes, as all lessons followed the provincial curriculum. Moreover, only the children with parental consent were assessed.

Finally, at the community level, five two-hour community nights were held in the school gymnasium where families could come with their children and be involved in different physical literacy opportunities with other community members. All families in the participating classes were invited to the community nights. Additionally, families were encouraged to bring their other children. The community nights were led by the facilitator and members of the research team (MCE, MLH). The goal of the community nights was to offer parents and children an opportunity to play together and develop physical competence. The community nights involved physical literacy activities, generally in the form of games that could be easily implemented by the participants outside of the organized setting. Unfortunately, compliance was not measured in any setting during the intervention.

To summarize, our intervention focused on a variety of skills, progressive challenges, collaboration with peers, individual achievement, and fun, which are key aspects for developing physical literacy [19,20,46,47].

### 2.4. Statistical Analysis

Descriptive statistics were run to describe the sample, including demographics such as grade and the number of males and females in each group. To evaluate the effectiveness of the intervention, a repeated measure MANOVA was used to assess the intervention effect pre- and post-intervention in each of the 10 skills, the domains, and overall physical competence. Separate repeated measures MANOVAs were used to assess the intervention effect on PLAY Parent and PLAY Coach values. All statistics were performed using SPSS version 28, and the *p*-value was set at 0.05.

## 3. Results

Two hundred and seventeen children from kindergarten and grade one participated and were measured at baseline; 31 children (15 intervention and 16 usual practice) were removed from the analysis because they did not have follow-up data. Thus, 186 children with complete pre- and post-intervention PLAY Fun assessments were included in the current analyses. PLAY Parent and PLAY Coach were obtained in one of the two cities and thus are presented as a sub-sample of the total study population. Table 1 displays the sample size breakdown for each PLAY tool. Post-testing data collection was conducted within 2 weeks of the conclusion of the intervention.

### 3.1. PLAY Fun

Differences in overall physical competence are presented in Figure 1. Overall physical competence improved for children in both the intervention (*p* < 0.000; 95% CI, 32.5–35.3) and usual practice groups (*p* < 0.000; 95% CI, 25.9–30.1). At follow-up, children in the intervention group (33.9 ± 6.6) had greater overall physical competence than children in the usual practice group (27.9 ± 9.5; *p* < 0.000; 95% CI, 32.5–35.3; ηp^2^ = 0.127).

#### 3.1.1. Motor Skill Comparison

PLAY Fun scores at baseline and follow-up for each skill are shown in Figure 2. Children in the intervention improved their physical competence for skip (*p* < 0.000; 95% CI, 36.6–42.5), gallop (*p* < 0.000; 95% CI, 40.4–45.4), hop (*p* = 0.001; 95% CI, 8.3–14.2), jump (*p* = 0.009; 05% CI, 14.2–19.1), throw (*p* < 0.000; 95% CI, 20.3–28.1), catch (*p* < 0.000; 95% CI, 25.0–32.3), kick (*p* < 0.000; 95% CI, 40.4–44.4), and balance (*p* < 0.000; 95% CI, 37.1–41.1). However, they displayed a decrease in their run there and back (*p* = 0.001; 95% CI, 37.3–46.4). Children in the usual practice group improved their jump (*p* = 0.002; 95% CI, 17.9–25.2), throw (*p* < 0.000; 95% CI, 26.8–33.1), catch (*p* < 0.000; 95% CI, 22.6–28.9), kick (*p* < 0.000; 95% CI, 30.7–35.8), and balance (*p* < 0.000; 95% CI, 29.3–34.2), while their run there and back (*p* = 0.002; 95% CI, 36.3–48.6) also decreased.

At follow-up, children in the intervention group had higher physical competence for skip (*p* < 0.000; ηp^2^ = 0.290), gallop (*p* < 0.000; ηp^2^ = 0.493), kick (*p* < 0.000; ηp^2^ = 0.190), and balance (*p* < 0.000; ηp^2^ = 0.111) (Figure 2). In contrast, the usual practice group had higher physical competence for balance at baseline (*p* = 0.003; ηp^2^ = 0.047) and greater jump (*p* = 0.033; ηp^2^ = 0.009) and throw (*p* = 0.037; ηp^2^ = 0.011) at follow-up.

#### 3.1.2. Domain Comparison

Figure 3 displays the domains for the PLAY Fun scores. Children in the intervention group decreased in the run domain (*p* = 0.010; 95% CI, 99.1–111.9) but improved in the locomotor (*p* < 0.001; 95% CI, 103.5–117.2) and object control domains (*p* < 0.001; 95% CI, 88.4–102.1). Children in the usual practice group demonstrated no change in the run domain (*p* = 0.089; 95% CI, 92.1–109.4) and improved their locomotor (*p* = 0.009; 95% CI, 50.2–61.6) and object control domains (*p* < 0.001; 95% CI, 82.3–95.7). No group differences were found in the run and object control domains. Children in the intervention scored higher in the locomotor domain post-intervention (*p* < 0.001).

### 3.2. PLAY Parent

PLAY Parent results are shown in Table 2. No differences were found for any of the PLAY Parent domains between intervention and usual practice children (*p* > 0.05).

### 3.3. PLAY Coach

PLAY Coach physical competence domain scores are presented in Table 3. Teachers of children in the usual practice group reported their students had greater balance competence (*p* = 0.001; ηp^2^ = 0.139), object control competence (*p* = 0.020; ηp^2^= 0.075), locomotor competence (*p* = 0.016; ηp^2^ = 0.080), and overall physical competence (*p* = 0.006; ηp^2^ = 0.104) at baseline. No group differences were found at follow-up (*p* > 0.05).

## 4. Discussion

The aim of this study was to assess if a 12-week multi-setting (home, school, and community) intervention could improve the physical competence of children in kindergarten and grade one. We found improvements in locomotor, object control, and overall physical competence for both students in the intervention and usual practice. However, children in the intervention group had greater locomotor and overall physical competence post-intervention. Surprisingly, children in the intervention decreased their physical competence in the running domain.

For specific skills, children in the intervention group significantly improved their skip, gallop, hop, jump, throw, catch, kick, and balance physical competence. Children in the usual practice group improved their jump, throw, catch, kick, and balance physical competence. Additionally, children in the intervention group had higher skip, gallop, kick, and balance physical competence, while the usual practice group had higher jump and throw physical competence post-intervention. The improvements observed for children in the usual practice group could be from involvement in sports that are developing those skills (i.e., jump—dance and gymnastics; throw and catch—T-ball and baseball; kick—soccer) or a greater focus on those skills by their teachers in physical education. There were no differences in parental perceptions of children’s physical competence between parents of children in the intervention and usual practice groups. Teachers of children in the intervention perceived that their students had lower physical competence compared to teachers in the usual practice group before the intervention, but no differences in teacher perception of physical competence existed after the intervention.

The improvements in physical competence are not surprising since physical literacy interventions have the strongest effect on physical competency outcomes [48]. When compared to the outcomes of other interventions utilizing the PLAY tools, we found similar improvements in physical competence for children in the intervention and, to a lesser extent, improvements for the usual practice group [31,32,35]. The percent change for overall physical competence in the current study was much higher for both the intervention and usual practice groups (44% and 14%, respectively), compared to other studies that found a smaller increase of 3% for the control group [32] and 8% for the intervention group [32,37]. One reason our percent change could have been higher is because our intervention focused on the motor development of fundamental movement skills rather than circus arts [32] or sport development [37] or potentially due to the multi-setting approach utilized.

For specific skills, similar results have been reported for gallop, hop, throw, catch [32], skip, kick [32,37], object control [31,37], and balance [30,37] improvements post-intervention in older children. Our finding of a decrease in physical competence to run there and back for the intervention group was surprising since running has been reported to improve post-intervention in other studies [30,32,37]. This finding could suggest the instructors and parents may have assumed children already knew how to run properly, and thus may not have emphasized their instruction on running development, or that running errors were not fixed.

The results from the PLAY Coach measurement indicated that teachers of children in the intervention group perceived their students’ physical competence as improving over time compared to teachers of children in the usual practice group. While this result is positive, because teachers of the intervention students were present for the physical literacy-enriched lessons and were asked to teach one of the lesson plans introduced that week to their class, they were aware of their students’ involvement in the intervention. This awareness could have influenced them to score these students more favorably after the intervention. However, teachers were not provided with the pre-scores they assessed their students; therefore, they would have had to remember what they scored their students’ 12-weeks previously. Additionally, given the fact that teachers in the intervention group scored their students lower at baseline compared to teachers in the usual practice group, the teachers in the intervention group may have perceived that their students lacked physical competence, and the intervention decreased the gap between the two groups.

The length of the intervention likely influenced our results; other interventions of a similar duration have also found benefits [31,49], suggesting that a 12-week intervention may be sufficient to create change. A systematic review found a negative, non-significant relationship between the effect size of pre- to post-intervention measurements of physical competence and intervention duration [50]. Unfortunately, due to the small sample of studies included, a critical point could not be identified [50]. Therefore, future research should examine the optimal duration for physical competence improvement. Furthermore, duration was calculated as total time rather than instructional time; therefore, intervention sessions could have involved greater instruction time and less time to practice motor skills, highlighting the need for clarity around the time allotted for the development of physical competence through practice.

The multi-setting approach used in the current study may be more beneficial than solely a school-based intervention by providing ample opportunities for children to practice these skills at school, at home, and at community nights. Additionally, by design, the intervention was non-competitive and non-comparative, which may have been advantageous as these environments have been suggested to make children more willing to engage in new motor skills [32]. Thus, it is likely the children in the intervention may have felt more comfortable trying new skills since they were active in multiple settings and not being compared to their peers. Finally, a systematic review found that characteristics of interventions that improve physical competence include collaboration between teachers and researchers for intervention implementation and parental involvement at home [51], all of which were key factors in the current study’s intervention design.

A limitation of the current study is the lack of compliance measures for the intervention in the home and community settings. While students returned their backpacks weekly to be given new equipment and physical literacy cards, and the facilitator reminded students to practice the new skills at home, we did not measure how often they were using the equipment and practicing at home. Having a compliance measure would have informed how often the children were engaging in motor development in each setting, providing more insight into how the different settings impacted physical competence. Another limitation was the need to adapt the PLAY Fun tool to be more age-appropriate. Removing skills deemed developmentally inappropriate could have affected the validity of the tool; however, the skills retained covered all the movement domains, and only the criteria for evaluating balance was altered, which may have helped to protect the integrity of the tool. Finally, since the PLAY Coach has not been validated, considerations need to be taken when interpreting the results.

Having multiple assessors could have led to interpersonal reliability issues; however, all assessors were well trained, and previous studies have reported that the PLAY Fun tool has good to excellent inter-rater reliability [43,44]. Additionally, the assessors evaluated the same skills at both baseline and follow-up. Even with limitations, PLAY Fun is a good tool for measuring physical competence [44]. A strength of our study is that one school division requested the researchers use passive consent, which likely increased our enrollment and gave us a more representative sample.

## 5. Conclusions

In conclusion, a 12-week multi-setting intervention was found to be effective at improving the physical competence of children in kindergarten and grade one. Teachers of children in the intervention group perceived these changes, while parents did not. We found significant improvements in physical competence for both children in the intervention and usual practice, with a significant intervention effect. Future studies should investigate the benefits of employing multi-setting physical literacy interventions. Since physically literate children should be able to engage in a variety of motor skills in different environments (e.g., ice, snow, and water) and settings (e.g., home, school, and community), conducting an intervention in one setting may not assist in the development of certain aspects of their physical literacy.

## Figures and Tables

**Figure 1 children-10-01364-f001:**
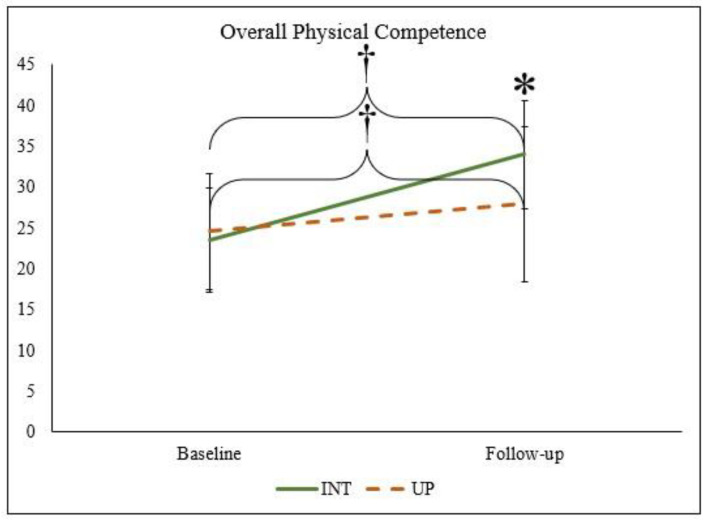
A paired *t*-test was completed to determine differences between pre- and post-intervention within each group. An independent *t*-test between intervention and usual practice at pre- and post-intervention was performed. † denotes a significant difference between the pre- and post-interventions (*p* < 0.000). * denotes a significant difference between the intervention and usual practice groups (*p* < 0.000). Abbreviations: Intervention (INT), Usual Practice (UP).

**Figure 2 children-10-01364-f002:**
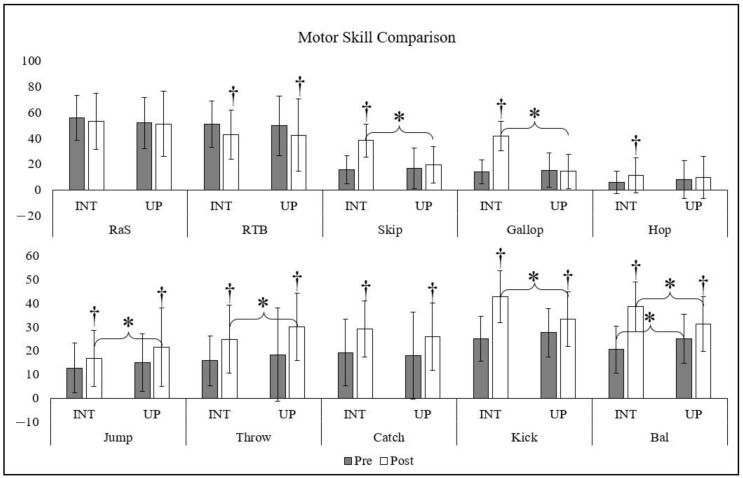
† denotes a significant difference between pre- and post-intervention within each group (*p* < 0.05). * denotes a significant difference between intervention and usual practices at either pre- or post-intervention (*p* < 0.05). Abbreviations: Intervention (INT), Usual Practice (UP), Pre-intervention (Pre), Post-intervention (Post), Run a Square (RaS), Run There and Back (RTB), Balance (Bal).

**Figure 3 children-10-01364-f003:**
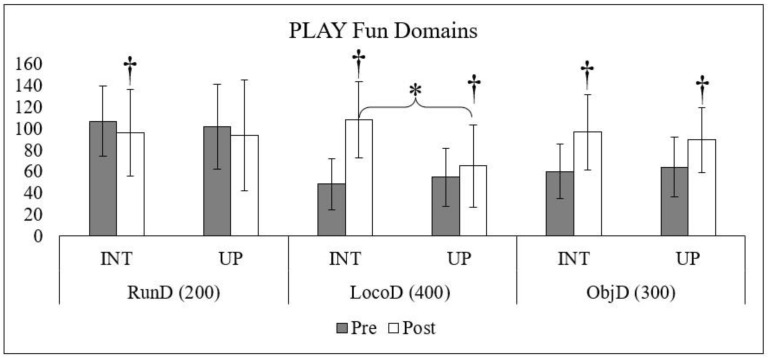
† denotes a significant difference between pre- and post-intervention within each group (*p* < 0.05). * denotes a significant difference between intervention and usual practices at either pre- or post-intervention (*p* < 0.05). The maximum score for each domain is listed beside the abbreviation in brackets. Abbreviations: Intervention (INT), Usual Practice (UP), Run Domain (RunD), Locomotor Domain (LocoD), Object Control Domain (ObjD).

**Table 1 children-10-01364-t001:** Participant sample size breakdown (n).

		Kindergarten	Grade 1	Total
**PLAY Fun**				
Intervention		48	55	103
	Male	31	31	62
	Female	17	24	41
Usual Practice		41	42	83
	Male	25	22	47
	Female	16	20	36
**PLAY Parent**				
Intervention		10	14	24
	Male	4	5	9
	Female	6	9	15
Usual Practice		10	16	26
	Male	7	8	15
	Female	3	8	11
**PLAY Coach**				
Intervention		20	21	41
	Male	12	9	21
	Female	8	12	20
Usual Practice		14	17	31
	Male	11	8	19
	Female	3	9	12

**Table 2 children-10-01364-t002:** PLAY Parent Domain Comparisons.

	Intervention			(95% CI)	Usual Practice			(95% CI)	*p*-Value
LocoD Pre	9.4	±	2.1	(8.6–10.5)	9.2	±	1.9	(8.4–10.1)	0.718
LocoD Post	9.4	±	1.8	(8.4–10.2)	9.7	±	1.9	(8.6–10.4)	0.611
ObjD Pre	4.8	±	1.1	(4.3–5.3)	4.3	±	1.5	(3.7–5.0)	0.282
ObjD Post	4.8	±	0.9	(4.3–5.3)	4.9	±	1.1	(4.4–5.4)	0.678

denotes a significant difference between intervention and usual practice groups at one time point. Abbreviations: Locomotor Domain (LocoD), Object Control Domain (ObjD).

**Table 3 children-10-01364-t003:** PLAY Coach Domain Comparisons.

	Intervention			(95% CI)	Usual Practice			(95% CI)	*p*-Value
BalD Pre	5.3	±	1.6	(4.3–5.6)	**6.7 ***	±	1.9	(5.8–7.6)	**0.001**
BalD Post	6.6	±	2.1	(4.9–7.2)	6.5	±	1.5	(5.8–7.4)	0.419
ObjD Pre	5.5	±	1.9	(4.2–6.1)	**6.5 ***	±	1.8	(5.7–7.3)	**0.020**
ObjD Post	6.6	±	2.1	(4.9–7.3	6.5	±	1.4	(5.7–7.2)	0.333
LocoD Pre	5.6	±	1.6	(4.2–5.8)	**6.7 ***	±	2.3	(5.7–7.7)	**0.016**
LocoD Post	6.8	±	2.3	(4.9–7.3)	6.6	±	1.7	(5.7–7.5)	0.361
PCD Pre	19.8	±	5.7	(15.8–21.1)	**24.0 ***	±	6.9	(21.1–27.2)	**0.006**
PCD Post	24.7	±	7.8	(18.3–26.7)	23.8	±	5.5	(21.1–26.8)	0.582

* denotes a significant difference between intervention and usual practice groups at one time point. Abbreviations: Balance Domain (BalD), Object Control Domain (ObjD), Locomotor Domain (LocoD), Physical Competence Domain (PCD).

## Data Availability

The data presented in this study are available upon reasonable request from the corresponding author.

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
