# Peer review of "Parents, Teachers, and Community: A Team Approach to Developing Physical Competence in Children"

_children, 2023, doi:10.3390/children10081364_

Round 1

Reviewer 1 Report

In recent years, "physical literacy" has received more and more attention in the field of sports. However, there is not enough research to explore effective ways to improve children's physical competence. This study aims to assess the tripartite physical literacy interventions of parents, teachers, and the community to improve children's physical competence. I think this research will contribute to the improvement of children's physical ability and even physical literacy. However, after reading this article, I still have some questions, which I need to ask the author to answer.

Introduction

1. Line 46-19: While stated logically, there is a lack of evidence of correctness. Please explain.

2.Paragraph 5: This paragraph points out the importance of school for the development of children's physical literacy. But that may be a little too one-sided. You can first explore how children's physical competence can be improved from more perspectives, such as playing with peers, and then explain why you want to focus on school.

3.Paragraph 4 and 5: According to the article, it seems that the more you participate in physical activity, the more your physical competence will improve. But how to prove that the amount of participation is related to the amount of field provided?

Materials and Methods

1. Line 117: there is a lack of evidence of between “PLAY Parent,” “PLAY Coach” and children competence. Please explain.

2.Paragraph 3: The researchers have made corrections for the measurement items of "PLAY Fun". What is the basis, and is this measurement method reliable and valid?

3.Paragraph 5: Does parental background, such as level of physical literacy, level of physical ability, level of education, etc., affect differences in the implementation of interventions? Also, parents who agreed to participate in this research have higher physical literacy themselves?

4.Intervention: Please describe more about the content of the intervention course and why it can improve physical literacy.

Results

1.Line 227: Since certain motor skill is required to throw, please discuss why the control group is better than the intervention group.

2. Is this study using random sampling? If so, will there be differences in the basis of the experimental subjects? What did you do to solve this problem?

3.Line 250: Since there are no differences were found for any of the PLAY Parent domains between intervention and usual practice children. Why you still suggest that providing ample opportunities for children to practice these skills at home in Line 328-330?
4.Line 255: Since there are no differences were found for any of the PLAY Coach domains between intervention and usual practice children. Could you discuss if PLAY Coach is not an effective way to improve children's physical ability.

Discussion

1.Paragraph 3: The author wrote that the data of this study has made great progress compared with previous studies, but from the results part, there is not much difference between the experimental group and the control group. please explain.

2. If PLAY Parent and PLAY Coach can only help distinguish children's physical competence, but cannot improve it, it is against the purpose of this research. Please explain.

3.What is the relationship between PLAY Parent and PLAY Coach and research intervention and children's physical competence development? Please explain.

Author Response

We would like to thank the reviewers for their comments and suggestions. We believe that in addressing your comments the manuscript has improved significantly.

REVIEWER 1:

Introduction

  1. Line 46-19: While stated logically, there is a lack of evidence of correctness. Please explain.

The end of the statement has been altered to “focusing on improving physical competence in the early years of life might have the greatest impact on children’s physical activity engagement [15-17]” as there is more evidence to suggest physical competence is beneficial for improving physical activity engagement throughout childhood, which is discussed further in paragraph three.

2.Paragraph 5: This paragraph points out the importance of school for the development of children's physical literacy. But that may be a little too one-sided. You can first explore how children's physical competence can be improved from more perspectives, such as playing with peers, and then explain why you want to focus on school.

We acknowledge that physical competence is developed in many contexts beyond the school and have added a paragraph to reflect this (now paragraph 5):

“Despite the importance of sport, 26% of Canadian children 5 to 17 years of age do not participate in organized sport [4], thus unstructured or free play is another essential way to develop physical competence. Free play can happen in any environment or setting, be with or without peers, use various types of equipment and requires no planning which allows children to have autonomy over the development of their physical competence. Free play has a positive influence on the development of locomotor physical competence and engagement in physical activity [24]. However, its’ impact on object control physical competence is limited by access to equipment [24]. Additionally, a review found that free play has the weakest impact on developing physical competence compared to structured opportunities like physical education classes and motor development interventions [25].”

At the start of paragraph 6 (formerly paragraph 5), we have added the statement “For children in kindergarten and grade one,” to indicate that the home and school are important settings for the age range of children in the current study. As children age it is likely the impact different settings have on children’s physical competence changes. 

3.Paragraph 4 and 5: According to the article, it seems that the more you participate in physical activity, the more your physical competence will improve. But how to prove that the amount of participation is related to the amount of field provided?

We apologize, we are unsure what the reviewer is requesting, is this referring to a specific article within one of the paragraphs or our submitted manuscript. Additionally, could you please clarify what is meant by “how to prove that the amount of participation is related to the amount of field provided”, in particular we are uncertain of what “the amount of field” means. 

 Materials and Methods

  1. Line 117: there is a lack of evidence of between “PLAY Parent,” “PLAY Coach” and children competence. Please explain.

PLAY Fun is a valid measurement of physical competence (Bremer et al., 2019; Caldwell et al., 2021; Cairney et al., 2018; Stearns et al., 2018) and has a moderate association to PLAY Parent (Caldwell et al., 2021). So, there is evidence to suggest PLAY Parent can effectively measure physical competence. We have added “PLAY Fun [42–45] and PLAY Parent [44] are valid measures of physical competence and have a moderate relationship [44]” to clarify.

Currently, there is a lack of evidence to suggest PLAY Coach can measure physical competence. To address this comment, we have added “The validity and relatability of PLAY Coach has not yet been studied. PLAY Fun evaluated the intervention’s impact on children’s physical competence while the PLAY Parent and PLAY Coach assessed if parents and teachers perceived any change in the children’s physical competence after the intervention” to paragraph 1 of section 2.2 Physical Competence Assessment. Additionally, “Finally, since the PLAY Coach has not been validated, considerations need to be taken when interpreting the results” has been added to the limitations section of the manuscript.

2.Paragraph 3: The researchers have made corrections for the measurement items of "PLAY Fun". What is the basis, and is this measurement method reliable and valid?

Thank you for the comment. Modifications occurred to make the tool more age appropriate as some of the original skills in the PLAY Fun did not align with the Saskatchewan physical education curriculum for kindergarten and grade one students. Thus, adaptation allowed the research team to ensure that the tasks were developmentally appropriate. We adjusted or modified two skills (two-handed catch and stationary balance), but the criteria for assessing skills was only altered for balance, to preserve the validity of the tool. The stationary balance utilized the criteria from an established stationary balance test that is used as part of the physical literacy passport of life assessment. We have added “The tool was not validated after the alterations. However, only the criteria for balance was modified from the original version created by Canadian Sport for Life [39] to account for the stationary balance test. The stationary balance test utilized the criteria developed for the physical literacy Passport for Life assessment tool [46]” to paragraph 2 of 2.2 Physical Competence Assessment as well as “and only the criteria for evaluating balance was altered,” to the limitations section.  

3.Paragraph 5: Does parental background, such as level of physical literacy, level of physical ability, level of education, etc., affect differences in the implementation of interventions? Also, parents who agreed to participate in this research have higher physical literacy themselves?

Parental background would affect their ability and willingness to participate in the intervention. To address this, we designed the home-based intervention instructional cards to for use by parents regardless of their background or level of physical literacy (Appendix 1). Additionally, the home aspect was only one part of the intervention. The students were also invited to participate in the physical education and community night aspects of the intervention; therefore, even if the home aspect was affected by parental background the school would not have been. This is part of the strength of a multi-setting approach.

It is possible that parents who consented for their children to participate had a higher understanding of or levels of physical literacy. However, one city used passive consent, thus it is likely we had a more representation sample since parents had to opt out of participating. In this community we had a greater than 90% participation rate from the classrooms, suggesting that all levels of students were represented.  

4.Intervention: Please describe more about the content of the intervention course and why it can improve physical literacy.

At home, the instructional cards were designed for parents to implement and improve their child’s physical competence, motivation, confidence, and comprehension. The cards contained information to help the parents play with their children and develop their child’s physical literacy. The following has been added into the manuscript: “The activity cards including information on how to play the activity, cues to provide the children, and the skills that were being developed as well as questions for parents to ask their children to help build their confidence and competence with the movement skills” An example of an instructional card will also be added as supplementary data (Appendix 1).

At school, lesson plans were developed to be used in physical education classes to develop the physical literacy of all students.  These lessons, developed by physical education specialists, included warm-up, skill development, and a culminating activity. The following has been added to the manuscript: The lesson plans identified the curricular outcome that would be achieved in the lesson as well as the equipment needed and a detailed explanation of 3-5 activities that would develop physical competence an example of a lesson plan has been added as supplementary data (Appendix 2).

At community, two authors led similar skill development and culminating activities as the school lesson plans. In both the school and community setting, the activities focused on a variety of physical activity opportunities that prioritized participation and exemplary pedagogy. For example, children were not standing in line waiting for a turn to participate but rather always actively engaged, modifications were made to each activity to enhance participation of children of a wide range of abilities. Furthermore, the lesson plans were included ways for the physical educator to increase or decrease the challenge of a particular activity to meet the developmental needs of each student. Finally, the community nights provided children the chance to develop skills with peers as well as on their own.

We have also added “To summarize, our intervention focused on a variety of skills, progressive challenges, collaboration with peers, individual achievement, and fun which are key aspects for developing physical literacy [20, 21, 47, 48]” to the very end of 2.3 Intervention to provide a overall description of how the intervention was designed to improve physical literacy.

Results

1.Line 227: Since certain motor skill is required to throw, please discuss why the control group is better than the intervention group.

This is an interesting finding, but one that is not limited to our study. We have added a sentence in the discussion to address this. “The improvements observed for children in the usual practice group could be from involvement in sports that are developing those skills (i.e., jump – dance and gymnastics, throw and catch – T-ball, kick – soccer) or a greater focus on those skills by their teachers in physical education.”

  1. Is this study using random sampling? If so, will there be differences in the basis of the experimental subjects? What did you do to solve this problem?

A repeated measures cluster randomized control trial was used to assess the impact of the intervention. The participating schools were matched for demographics, programming, and social supports as well as interest in the present study. Therefore, there should not be any differences in the students based on demographics and programming opportunities such sport availability. Randomization occurred at the school level to minimize potential of cross-contamination; so that students receiving the intervention would not influence the students receiving the usual practice physical education programming. We have added “using a repeated measures cluster randomized control trial” to paragraph 1 of 2.1 Research Design and Participants.

3.Line 250: Since there are no differences were found for any of the PLAY Parent domains between intervention and usual practice children. Why you still suggest that providing ample opportunities for children to practice these skills at home in Line 328-330?

It is important to note that while parents did not perceive that their children had a greater improvement in physical literacy the researcher assessed PLAYfun and teacher assessed PLAY coach did find improvements in physical competence. Therefore, the home-based portion was important for improving the child’s physical competence (PLAY Fun scores) since the repeated practice of skills at home, that were also practiced at the school and community may have amplified the impact of the intervention. So while the parents did not perceive an improvement it does not mean that the home was not an important site for skill development.

4.Line 255: Since there are no differences were found for any of the PLAY Coach domains between intervention and usual practice children. Could you discuss if PLAY Coach is not an effective way to improve children's physical ability.

The PLAY Coach was used to assess the change in the teacher perspective of children’s physical competence rather than improve their physical ability. The teachers were provided with the lesson plans in the intervention school to improve the physical literacy development of their students. The PLAY coach is a tool to assess physical literacy and physical competence at a given time point it is not designed to affect change. While there were no differences in the PLAY coach between the intervention and usual practice. Teachers in the intervention school did perceive that their students had a greater improvement in physical competence post intervention compared to the usual practice.

 Discussion

1.Paragraph 3: The author wrote that the data of this study has made great progress compared with previous studies, but from the results part, there is not much difference between the experimental group and the control group. please explain.

In lines 320-323 we state that the results of this study found “found similar improvements in physical competence for children in the intervention and, to a lesser extent, improvements for the usual practice group [32, 33, 36].” We go on to state that the percent change (from baseline to post intervention) was much higher in the current study compared other studies. However, it was not our intention to suggest that the data of this study has made “great progress compared to others”. However, when looking at overall competence (Figure 1) the percent change in the intervention group is 44% compared to the usual practice at 14%. We would suggest this is fairly large difference and we also note that our findings display a greater change over time than the than other published interventions (8%). We on go to state that this difference might be due to: “our intervention focusing on the motor development of fundamental movement skills rather than circus arts [33] or sport development [38] or potentially due to the multi-setting approach utilized.” Compared the studies that report an ~8% change in the intervention group in overall physical competence.

  1. If PLAY Parent and PLAY Coach can only help distinguish children's physical competence, but cannot improve it, it is against the purpose of this research. Please explain.

PLAY Parent and PLAY Coach tools were used to evaluate the change over time in the parent and teacher perspective of the children’s physical competence. These tools are not designed to improve physical competence but rather measure if parents and teachers perceived physical literacy intervention was successful at improving physical competence. These tools are commonly used in the area to assess the effect of an intervention designed to improve physical competence or physical literacy (Hennessy et al., 2018; Kozera, 2017; Kriellaars et al., 2019).

Our intervention included instructional cards at home, physical literacy lesson plans developed for the physical education classes combined with the community nights to improve physical competence. The PLAY tools (PLAY Fun, PLAY Parent, and PLAY Coach) were then used to assess if the intervention components at the home, school and community were effective at improving physical competence.

3.What is the relationship between PLAY Parent and PLAY Coach and research intervention and children's physical competence development? Please explain.

The research intervention (consisting of the home, school and community components described in lines 171-214 of the manuscript) was designed to improve children’s physical competence, while the PLAY Parent and PLAY Coach were used to measure if parents and teachers perceived any change in the children’s physical competence. The following has been added to the first paragraph of 2.2 Physical Competence Assessment “PLAY Fun evaluated the intervention’s impact on children’s physical competence while the PLAY Parent and PLAY Coach assessed if parents and teachers perceived any change in the children’s physical competence” to add clarity.

Reviewer 2 Report

This work is based on a scientific research intervention strategy with a 3-month action period combining school, family and community. This combination of actions is most effective if it enables children to improve their motivation and personal choices.

Running poses a problem under the age of 9, as the postural system is not mature enough for this activity. Around age 7, the motor activity is to walk on a 5 cm beam without falling, and at age 8 to throw a ball at a target. Have the exercises been adapted to the maturation of the postural system?

The effects of the intervention strategies last only a few weeks after the interventions. Can the authors specify how long after the intervention the tests were carried out?

Environmental modifications are an important element in the development of the postural system. Have such modifications been suggested?

Author Response

We would like to thank the reviewers for their comments and suggestions. We believe that in addressing your comments the manuscript has improved significantly.

REVIEW 2:

Running poses a problem under the age of 9, as the postural system is not mature enough for this activity. Around age 7, the motor activity is to walk on a 5 cm beam without falling, and at age 8 to throw a ball at a target. Have the exercises been adapted to the maturation of the postural system?

As you have noted, postural control is a limiting factor in physical competence, particularly in the first years of life (Clark et al., 2007). For children in our study, although posture may not be full matured, running is a common fundamental motor skill that children should be engaging in (Canadian Sport for Life, 2019; Clark & Metcalfe, 2002; Hulteen et al., 2018; Newell, 2020; Seefeldt, 1980).

Since PLAY Fun tool is designed to monitor children’s physical competence development maturational adaptations were not made as the running criterion denotes a gradient from initial characteristics such as ‘immature running form’ to ‘mature running form’ as a key characteristics of a proficient running classification. In other words, the PLAY Fun acknowledges children have to improve their postural control to be competent runners. As shown in Appendix 1, the tool classifies individuals into 4 quadrants that each represent a different level of motor skill competence. For the children in our study, we would expect them to display characteristics as described in the initial and emerging stages. Conversely, adolescents and young adults, who have a mature postural system, would likely display the “Mature running form evident” characteristic in the competent quadrant for running physical competence (Appendix 1). Thus, making maturational adaption would alter the validity of the tool. Motor skills were only removed from the PLAY Fun if children were not normally exposed to those tasks (i.e. dribble a basketball and over hand throw) at this young age and if they were not present in the Saskatchewan physical education curriculum for kindergarten and grade one.         

The effects of the intervention strategies last only a few weeks after the interventions. Can the authors specify how long after the intervention the tests were carried out?

The post-testing was completed within 2 weeks of conclusion of the intervention. At the beginning of the results section, we “Post-testing data collection was conducted within 2 weeks of conclusion of the intervention.” However, some studies have found that some intervention effects last after the completion of the intervention (Barnett et al., 2009; Lai et al., 2014; Zask et al., 2015) and it is our hope that these benefits would be maintained longer term.

Environmental modifications are an important element in the development of the postural system. Have such modifications been suggested?

One of the focuses of physical literacy is movement in different environment (i.e snow, water, ice). For this particular study the intervention was limited by the school context, as the majority of movement happened in a gymnasium. The at home activities focused on both indoor and outdoor environments as well as examples were given for both summer and winter seasons (Appendix 2). We have added “in both the winter and summer environments” to section 2.3 Intervention to address this comment. 

Appendix 1 – PLAY Fun Criterion Examples

Appendix 2 – Instructional Card

© University of Saskatchewan Physical Literacy Team (USPLIT)  

Round 2

Reviewer 1 Report

Dear authors,

Thank you for referring to my opinion and correcting the content. 
I think this is a great research, It will be very helpful for physical literacy development. In the future, you may be able to continue based on the unfinished part of this research.

Sincerely,
LIU